# Association between Having Cancer and Psychological Distress among Family Caregivers Using Three Years of a Nationwide Survey Data in Japan

**DOI:** 10.3390/ijerph181910479

**Published:** 2021-10-06

**Authors:** María Lisseth Morales Aliaga, Tomoko Ito, Takehiro Sugiyama, Timothy Bolt, Nanako Tamiya

**Affiliations:** 1Department of Health Services Research, Graduate School of Comprehensive Human Sciences, University of Tsukuba, Ibaraki 305-8575, Japan; marialisseth1987@yahoo.com; 2Health Services Research & Development Center, University of Tsukuba, Ibaraki 305-8575, Japan; tito@md.tsukuba.ac.jp (T.I.); tbolt@mail.saitama-u.ac.jp (T.B.); ntamiya@md.tsukuba.ac.jp (N.T.); 3Department of Health Services Research, Faculty of Medicine, University of Tsukuba, Ibaraki 305-8575, Japan; 4Diabetes and Metabolism Information Center, Research Institute, National Center for Global Health and Medicine, Tokyo 162-8655, Japan; 5Institute for Global Health Policy Research, Bureau of International Health Cooperation, National Center for Global Health and Medicine, Tokyo 162-8655, Japan; 6Department of Economics, Saitama University, Saitama 338-8570, Japan

**Keywords:** cancer, family caregivers, informal care, patients, psychological distress

## Abstract

We aimed to describe the characteristics of caregivers with cancer compared to those without and analyze the association between having cancer and caregivers’ psychological distress in Japan. We used data from the Japanese Comprehensive Survey of Living Conditions in 2010, 2013, and 2016. The participants were 5258 family caregivers aged ≥40 years, caring for only one family member whose information in the dataset was available for all the covariates included in the model. The family caregivers’ psychological distress was defined by the Kessler Psychological Distress Scale (K6) score (K6 ≥ 5). We conducted a Poisson regression analysis to examine the association between having cancer and family caregivers’ distress. The sample of family caregivers consisted of mostly females (69.3%) and people within the 40–64 years age group (51.8%). As a result, family caregivers with cancer increased across the survey periods; a higher number of participants were unemployed. When adjusted for covariates, including the presence of other diseases, having cancer was significantly associated with distress (risk ratio 1.33, 95% confidence interval 1.05–1.69) among family caregivers. Family caregivers with cancer are expected to increase in the future; it is important to provide them with more support in managing both their treatment and caregiving to cope with their distress.

## 1. Introduction

People live longer with more chronic conditions or other disabilities [1], and the need for long-term care is ever increasing. In Japan, long-term care insurance (LTCI) started in 2000 and has assisted aged persons with disabilities. Although services have been attempted to reduce the burden on family caregivers indirectly, they are still the preferred individuals to care for older adults [2]. There is concern about many issues related to family caregiving [2], including the aging of family caregivers [3,4,5].

As family caregivers age, their health becomes an important issue for the family caregivers themselves and the whole caregiving process [6]. A study in Japan reported that 50% of family caregivers presented with chronic illness [7]. Findings from a study in Hong Kong show that family caregivers have a higher risk of having hypertension, bronchitis, digestive ulcers, arthritis, and osteoporosis than non-family caregivers [8]. Psychological distress is also a well-known health problem among family caregivers, as reported that it is related to family caregivers’ and care recipients’ characteristics. As for care recipients’ characteristics, the association between care recipients’ disease and family caregivers’ distress has been well examined by many researchers, specifically care recipients’ cognitive dysfunction, with Alzheimer’s disease being identified as a strong factor of stress for family caregivers [9,10]. However, within the family caregivers’ characteristics, the association between their disease and distress, anxiety, or depression has only been mentioned in a few studies [11,12,13,14]. In Japan, while a nationwide survey reported that 33.0% of male and 27.1% of female family caregivers worried about their illness [15], the impact of specific kinds of diseases has not been discussed as a factor of family caregivers’ distress.

On the other hand, cancer is one of the primary causes of death among the general population in Japan, with middle-aged or older adults being more susceptible [16]. Cancer might influence their quality of life by negatively affecting their physical and psychological health. Previous studies among the general population in Asian countries showed that those having cancer presented higher distress than those without cancer [17,18]. A study from the USA reported an 8.5% prevalence of cancer amongst family caregivers [19], and family caregivers who had cancer were not unusual from the clinical staff perspective [20]. In recent years in Japan, given the concerns mentioned above about the increase in both cancer prevalence [16] and family caregivers’ age [15], the family caregivers who have cancer can be suspected to be of the most concern and have major issues.

The family caregivers with cancer might be considered as having difficulties in their double role as family caregivers and cancer patients. They are expected to feel highly distressed, and it is important to discuss additional interventions for them. Related to the distress among family caregivers with cancer, two previous studies outside of Japan have reported similar results of high distress among them [21,22]. One of the studies was from England [21] with a sample of cancer survivors, where being a family caregiver was a factor in the model for explaining the presence of distress in these survivors, with 20.6% of cancer survivors being family caregivers.

In another study from the USA [22], participants were recruited from medical facilities using community announcements and showed an unadjusted presentation of results. The use of a national database might have offered a generalizable representation of family caregivers with cancer. The Hassles and Uplifts Scales applied for this study measured the quantity and balance between negative demands and positive experiences, where the terms distress and stress are used interchangeably. The Hassles and Uplifts Scales are designed to measure stress, while the Symptom Distress Scale is designed to measure distress. Many terms that are used in both scales have similar meanings but measure different concepts [23]. Although distress is studied as an indicator of measuring mental health at a country-level for epidemiology, public health, and as an outcome in clinical trials and intervention studies, there is still confusion about its definition, as it is frequently and interchangeably used with other concepts such as strain, stress, etc. [23,24]. The issue of ambiguous terminology does not allow consensus, comparability, or generalizability of findings within the scientific community that is studying the same phenomenon. Furthermore, The Hassles and Uplifts Scales have not been utilized at a national level.

Moreover, both previously mentioned studies showed an unadjusted presentation of results since adjusting by covariates can control important variables that might affect a study’s outcome.

In addition, we found that in Japan, Kumagai studied the impact of high-intensity caregiving on family caregivers’ mental health (family caregiver distress; Kessler Psychological Distress Scale, K6), stratifying the sample by working status. This study intended to include family caregivers’ diseases as covariates and reported the differences in K6, although the effects of cancer, specifically, were not reported [25]. To our knowledge, there have been very few studies about the distress among family caregivers with cancer.

Therefore, we focused on the association between cancer and distress among family caregivers. This study aimed to describe the characteristics of family caregivers with cancer compared to those without and analyze the association between having cancer and family caregivers’ psychological distress in Japan.

## 2. Materials and Methods

### 2.1. Data Source

This is a cross-sectional study using data from the Comprehensive Survey of Living Conditions (CSLC), a Japanese nationwide and cross-sectional survey for households and household members’ living situations conducted by the Ministry of Health, Labour, and Welfare. The survey has been conducted annually, whereas questionnaires for long-term care recipients have been completed every three years. We used data from 2010, 2013, and 2016; samples were extracted independently in every survey. All questionnaires were self-administered (answered by a household member), with surveyors visiting households to distribute and collect the questionnaires.

The CSLC questionnaire collected information on the household itself, health, and long-term care. The response rate was 79.4%, 79.6%, and 77.6%, respectively. The long-term care questionnaire was administered only to the household members with a LTCI certification, whereas the household and health questionnaires were administered to all households.

### 2.2. Participants

The participants were primary/family caregivers who had an LTCI certification and provided care to aged, chronically ill, or disabled family members at home. In the CSLC, the care recipients were defined as respondents to the long-term care survey. The CSLC also included the information for the household of the respondent and for the family members who take care of the care recipients. With the information, we defined the family caregivers who take care of the care recipients as the primary study participants. This algorithm to define the family caregiver is used in previous studies using the CSLC [26,27]. We included participants aged 40 years or older because there were very few younger caregivers with cancer [28] and it is also the target age for cancer screening in Japan [29]. Additionally, participants caring for only one family member with information available in all relevant variables of the model and diseases associated with high distress were included. Then, family caregivers were matched with care recipients.

### 2.3. Dependent Variable (Outcome)

Distress was measured using the K6 scale, which was included in the CSLC and was self-administered. The K6 is a short survey with solid psychometric properties, widely used in general-purpose health surveys in the USA, Canada, Australia [30], and Japan [31]. It had been translated into Japanese and had shown acceptable internal consistency, reliability, and validity [31]. Its use has been validated in many studies, compared with other surveys, demonstrating the ability to discriminate anxiety and mood disorder cases from non-cases in the community following the Diagnostic and Statistical Manual of Mental Disorders-IV (DSM-IV) [32].

It included six items using a five-point response scale ranging from 0 (none of the time) to 4 (all of the time). Total scores ranged from 0 to 24, with a higher score indicating greater distress. Following previous studies [33], we treated the K6 scores as binary: 0–4 (normal) as no or low presence of distress and 5–24 as having psychological distress (moderate to severe distress). This cutoff point for the K6 has been probed for screening mood and anxiety disorders with a sensitivity of 100% and a specificity of 68.7% [34].

### 2.4. Independent Variable (Exposure)

The primary exposure for this study was cancer among family caregivers. The presence of disease among family caregivers was obtained from family caregivers having to receive outpatient care regularly. To identify whether a respondent had cancer or not, we created dichotomous variables using the family caregivers’ multiple answers from the diseases listed in the survey, including cancer. Therefore, having cancer and other diseases were not mutually exclusive.

### 2.5. Covariates

Having other diseases that were not mutually exclusive with each other were treated as covariates in the multiple regression analysis. These were considered as significant diseases related to high distress in the general population in previous studies, including angina/myocardial infarction [35], arthropathy [36], asthma [36], diabetes [36,37], eye disease [35,36], fracture [35], gastroduodenal disease [36,37], hypertension [35,36,37], liver disease [38], lower back pain [36], rheumatoid arthritis [36], osteoporosis [36], and stroke [35]. Cancer was also a disease associated with high distress [35]. Family caregivers who did not use outpatient care were categorized as non-diseased for all diseases in the definition of the dichotomous variables.

We collected the other covariates based on a literature review and their availability in our database. The covariate information was collected from both the family caregivers and their corresponding care recipients. The family caregiver variables included sex, age, educational history, job status, monthly expenditure per person, current smoking status, relationship to the care recipient, help from other family caregivers, and use of formal help care services.

The care recipient variables were sex, age, primary disease for care needs, support and need care level, and time spent caring.

Long-term care services in Japan are categorized as either requiring support or long-term care and are classified into seven levels. Support levels 1 and 2 offer services that mainly focus on the prevention of disability. The five levels within “requiring long-term care” range from care-need level 1 for users who are less disabled to care-need level 5 for users who are the most disabled [4,39].

### 2.6. Data Analysis

Descriptive analyses of the family caregivers with cancer characteristics were conducted. The subjects were stratified by cancer presence, and a bivariate analysis was performed using a chi-square test. To examine the association between cancer and distress among family caregivers, the distribution of K6 scores was stratified by cancer presence and was drawn using histograms. The association between having cancer and distress was adjusted for covariates and examined using the modified Poisson regression analysis with robust standard errors [40].

STATA Statistical Software release 15 (Stata Corp LLC, College Station, TX, USA, 2017) was used for all analyses.

### 2.7. Ethical Approval

The Medical Ethics Review Board approved this study from the University of Tsukuba (approval number: 1324). This study was retrospective in design and had no data identifying individuals. Therefore, informed consent and opt-out were impossible. The methods were based on ethical guidelines for research.

## 3. Results

In 2010, the long-term care validity questionnaires included 5912 persons requiring care. Additionally, data from 6342 people in 2013 and 6790 people in 2016 were collected (Figure 1). The surveyed participants were different each year, and the data for the surveys in 2010, 2013, and 2016 were appended to form a single dataset. The final participants were 5258 family caregivers (*n* = 1439 in 2010, *n* = 1948 in 2013, and *n* = 1871 in 2016) (shown in Table 1).

The share of family caregivers with cancer showed a slight increase through the three surveys (25.6% in 2010, 33.3% in 2013, and 41.1% in 2016). Most family caregivers with cancer were unemployed (72.2%) and higher in the cancer sample than the total sample mean (57.7%). Family caregivers with cancer smoked less than the full family caregivers’ sample (3.3% versus 13.2%). Details about the care recipients are provided in Appendix A.

Figure 2 below shows the distribution of distress scores (K6) within the cancer group and non-cancer group by percentage. In the non-cancer sample, 29.3% presented no distress (K6 = 0), whereas the proportion in the cancer group was 17.8%. A larger sample of the cancer group was distributed from scores 5 to 24 (moderate to severe distress) than the non-cancer group.

The Poisson regression analysis for the association between being a cancer patient and distress among family caregivers is shown in Table 2. In the model, adjusted with covariates including the presence of other diseases, cancer (risk ratio 1.33, 95.0% confidence interval 1.05–1.69) was related to distress significantly. More details of the covariates are provided in Appendix A.

## 4. Discussion

Being a cancer patient was significantly associated with more psychological distress among family caregivers in the multivariable model adjusted for both family caregivers’ and care recipients’ characteristics. This is the first study to illustrate the higher risk of psychological distress among family caregivers with cancer compared to those without cancer in Japan. Two previous studies outside of Japan have also reported similar results of high distress in family caregivers. In one study [21], a bivariate analysis found that cancer survivors who reported being a family caregiver presented more significant (*p* < 0.001) social distress (SD) (17.3%) than those who were not family caregivers (14.5%); this was confirmed by logistic regression analysis (OR 1.30, *p* < 0.001). In Vitaliano’s study [22], the family caregivers with cancer presented fewer uplifts (positive experiences) at the first and second measurement times and more hassles and uplifts in both times than family caregivers without cancer; as shown in a table, more family caregivers with cancer histories presented high stress (65.0%, *n* = 15) than low perceived stress (35.0%, *n* = 8).

These findings have important implications for developing additional interventions, specifically for family caregivers with cancer who need to manage their situation and, consequently, their distress. Otherwise, as the family caregiver’s morbidity or health declines, it can affect the care recipient’s health and recovery [41]. Family caregivers who are highly stressed neglect their own health due to taking care of the care recipient [42]. Similarly, it has been shown that care recipients who have worse well-being measures have significantly greater probabilities of visiting an emergency room [43]. According to other countries’ previous experience, the following measures can be undertaken through the family caregivers’ practitioners, medical staff, or the care recipients within the LTCI system in Japan.

A systematic process of gathering information about the caregiving situation may be useful to identify a family caregiver’s health, needs, strengths, preferences, and resources [41]. According to the guiding principles and practice guidelines from the National Consensus Development Conference for Caregiver Assessment in the USA, this assessment can be performed by the family caregiver’s physician or by other healthcare team members, including social workers, family caregivers, or the care recipient’s case manager. Family caregivers’ evaluations should also include perceptions of their well-being, challenges and benefits they perceive from caregiving, level of confidence in their skills, and the need for additional support systems. The assessment findings can be used to develop a care plan/program and identify appropriate support services [44], as hospitals can coordinate and manage both family caregivers and care recipients with their treatment or caregiving. They can also facilitate respite care and caregiving by other family members, friends, etc. We emphasize the importance of supporting family caregivers with cancer because of the previously stated burden and situation.

For academia, this study will uncover critical sub-areas in psycho-oncology that researchers have not explored yet. This might open new areas of study within this field that may continue to become more important due to increases in aging, cancer prevalence, and family caregivers as preferred in caregivers in the population. Our findings focusing on family caregivers having cancer will bring new study variables into psycho-oncology involving treating the patient with cancer, their family, and their family caregiver’s psychological responses to cancer at all stages of the disease as well as the psychological, behavioral, and social factors that might influence the disease process. Since the definition of distress—the presence of anxiety and depression—is the same in both psycho-oncology and this study, it may be able to be used for future research and for future worldwide implications. Since Japan is considered a super-aged society with the largest percentage of older adults (28%) in their population [45], the country is taken as a model of research and management of aging across different fields. Therefore, our study would bring important information about the psychological problems among family caregivers with cancer for clinicians, policy makers, and researchers in other countries.

The present study has several strengths that are also novel aspects in the study of caregiving with cancer and distress. Firstly, the CSLC, a nationwide survey, can represent the Japanese population. Second, this study used the K6 scale, a short, widely used scale with strong psychometric properties. Lastly, this study offers a more complete explanation of distress among family caregivers with cancer; other studies did unadjusted presentations of results [22], whereas our model is adjusted for covariates and controls for different factors to family caregivers’ distress.

This study has some limitations. First, this study had a cross-sectional design; therefore, we could not determine a causal relationship between family caregivers’ cancer and distress in our study population. A meta-analysis showed that various studies mentioned that there may be a negative feedback loop between a decline in psychological health and a decline in physical health [46], where worsening physical health is a risk factor for family caregiver’s depression [46,47]. In the future, we hope that this study can be expanded upon using a longitudinal dataset. Second, since the presence, or absence, of any disease among family caregivers was self-reported and not reported by a physician, we cannot be assured that it is accurate. Third, we did not show results for non-family caregivers because it surpasses our study’s aim. As previously reported, people with cancer present higher distress than those without cancer in the general population [17,18,35]. Therefore, future studies can investigate whether the relationship between distress and cancer can be modified by caregiving status. Fourth, we could not consider other health behavior variables besides smoking status, as drinking status and exercise were not accounted as covariates due to the limitations of the database. Fifth, there is no way to tell if the same family caregiver–care recipient dyads took part in more than one wave of cross-sectional surveys since our study data was anonymized. The inter-person correlations should have been statistically adjusted in our analyses if the same subjects participated in the survey multiple times, but we were unable to do so due to the lack of a subject identifier in our data. However, it is assumed to be very rare to participate in the survey multiple years in a row because each survey is extracted from about 2500 unit areas of approximately 1 million unit areas throughout Japan using a stratified sampling method [48,49]. Last, nonresponse bias might exist. A family caregiver with a high K6 score, which indicates the presence of anxiety or depression, might not be able to complete the survey; however, those who have no or less distress are more likely to answer due to their adequate mental status. Nevertheless, any such nonresponse bias among distressed survey respondents would be expected to be proportionate across the groups studied. Overall, this study makes important contributions to future research.

## 5. Conclusions

Family caregivers with cancer presented with higher distress than the ones without cancer. As expected, the prevalence of cancer is also increasing in family caregivers. Consequently, it is important to evaluate family caregivers with cancer and provide them with more support from hospitals to manage their treatment and caregiving, such as respite, a network of secondary family caregivers, or other measures. Our findings provide useful evidence to recognize that family caregivers with cancer have higher levels of distress and may assess hospitals for future support. However, research related to distress in family caregivers with cancer is still in development.

## Figures and Tables

**Figure 1 ijerph-18-10479-f001:**
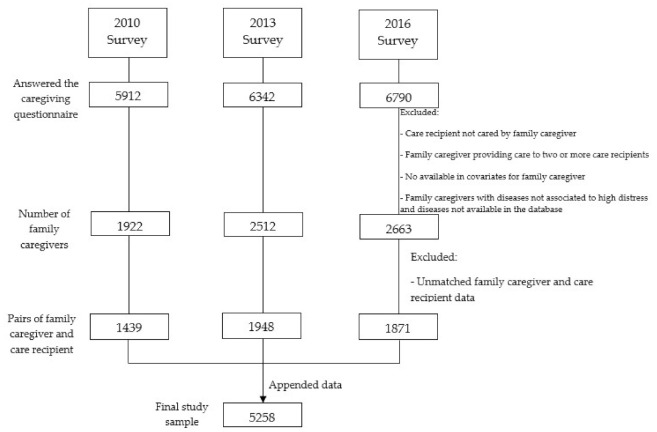
Sample flow chart.

**Figure 2 ijerph-18-10479-f002:**
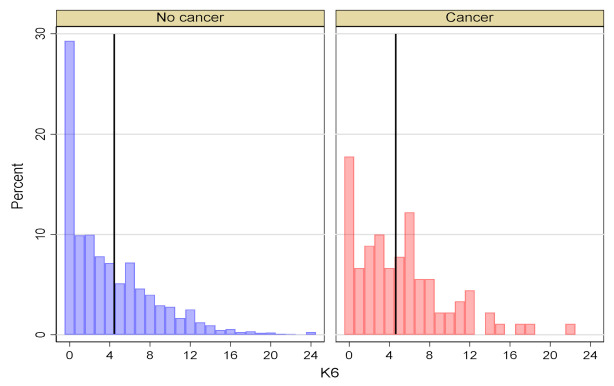
The proportion of family caregivers’ K6 scores within cancer presence groups.

**Table 1 ijerph-18-10479-t001:** Family caregivers’ characteristics by cancer presence.

		Total	Cancer	No Cancer	
		*n* (%) ^‡^	*n* (%) ^‡^	*n* (%) ^‡^	*p*-Value ^†^
Total		5258 (100.0)	90 (1.7)	5168 (98.3)	
Survey year	2010	1439 (27.4)	23 (25.6)	1416 (27.4)	0.54
	2013	1948 (37.1)	30 (33.3)	1918 (37.1)	
	2016	1871 (35.6)	37 (41.1)	1834 (35.5)	
Family caregivers’ Characteristics					
Sex	Male	1616 (30.7)	25 (27.8)	1591 (30.8)	0.54
	Female	3642 (69.3)	65 (72.2)	3577 (69.2)	
Age	40–64	2724 (51.8)	38 (42.2)	2686 (52.0)	0.24
	65–74	1288 (24.5)	29 (32.2)	1259 (24.4)	
	75–84	972 (18.5)	17 (18.9)	955 (18.5)	
	85+	274 (5.2)	6 (6.7)	268 (5.2)	
Education history	Elementary/junior high school	1429 (27.2)	18 (20.0)	1411 (27.3)	0.11
	High school	2533 (48.2)	53 (58.9)	2480 (48.0)	
	University/graduate school	1296 (24.7)	19 (21.1)	1277 (24.7)	
Job status	Having any job	2224 (42.3)	25 (27.8)	2199 (42.6)	0.005
	Not having a job	3034 (57.7)	65 (72.2)	2969 (57.5)	
Monthly expenditure ^§^	Less than 7.5 (×10,000 yen) per month	2605 (49.5)	40 (44.4)	2565 (49.6)	0.33
	From 7.5 (×10,000 yen) per month	2653 (50.6)	50 (55.6)	2603 (50.4)	
Current smoking status	Yes	696 (13.2)	3 (3.3)	693 (13.4)	0.005
	No	4562 (86.8)	87 (96.7)	4475 (86.6)	
Relationship with the care recipient	Spouse	1920 (36.5)	31 (34.4)	1889 (36.5)	0.61
	Son/daughter	1872 (35.6)	36 (40.0)	1836 (35.5)	
	Son-in-law/daughter-in-law	1331 (25.3)	19 (21.1)	1312 (25.4)	
	Parent	25 (0.5)	1 (1.1)	24 (0.5)	
	Other relatives	110 (2.1)	3 (3.3)	107 (2.1)	
Help from family members	Having help from family members	2552 (48.5)	43 (47.8)	2509 (48.6)	0.89
	Not having help from family members	2706 (51.5)	47 (52.2)	2659 (51.5)	
Use of formal help services	Using any formal help services	3357 (63.9)	59 (65.6)	3298 (63.8)	0.73
	Not using any formal help services	1901 (36.1)	31 (34.4)	1870 (36.2)	

Note: ^†^ Determined by chi-square test. ^‡^ Percentages do not sum to 100% due to rounding. ^§^ Monthly expenditure per person Low/Middle: <75,000 JPY per month. Monthly expenditure High: ≥75,000 JPY per month. (100 JPY = $1.00 American dollar approx.). Care recipients’ characteristics are listed in Appendix A.

**Table 2 ijerph-18-10479-t002:** Association between having cancer and distress among family caregivers.

	Adjusted ^†^		
Variable	RR	95% CI	*p*-Value
Cancer	1.33	1.05–1.69	0.016

Abbreviations: RR, risk ratio. CI, confidence interval. ^†^ Poisson regression analysis was adjusted for family caregivers’ variables, including age, sex, educational history, job status, monthly expenditure per person, smoking status, relationship to the care recipient, help from other family caregivers, and use of help care services. Family caregivers’ diseases included angina/myocardial infarction, arthropathy, asthma, diabetes, eye disease, fracture, gastroduodenal disease, hypertension, liver disease, lower back pain, rheumatoid arthritis, osteoporosis, and stroke. Care recipients’ variables included age, sex, primary disease for care need, support and care need level, and time spent caring. Additional family caregivers’ and care recipients’ variables in the model are listed in Appendix A.

## Data Availability

Because this is a secondary study using a national survey data, data sharing is not applicable.

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
