# Peer review of "Association between Having Cancer and Psychological Distress among Family Caregivers Using Three Years of a Nationwide Survey Data in Japan"

_ijerph, 2021, doi:10.3390/ijerph181910479_

Round 1

Reviewer 1 Report

I would like to thank the editor for offering this opportunity to review this manuscript. The purpose of this study is “ to describe the characteristics of caregivers with cancer compared to those without and analyze the association of having cancer and caregivers’ psychological distress in Japan”. The manuscript is generally well written; reflects rigorous methodology; and contributes to the association between having cancer and psychological distress among caregivers. My comments to enhance the quality of the manuscript are few.

  1. Caregivers and family caregivers are not exactly the same, I would consistently use one of them, e.g., family caregivers, throughout the whole manuscript.
  2. I am confused by the description of hypothesis. It seems to me that the descripted hypothesis is a common sense in that: “having cancer is expected to be related to higher distress among caregivers.”, which may limit the novelty of the manuscript.
  3. I am also confused by the description of the study purpose in introduction section (page3, lines 100-102), maybe it is easier to understand as presented under abstract section: “to describe the characteristics of caregivers with cancer compared to those without and analyze the association of having cancer and caregivers’ psychological distress in Japan.”
  4. Considering the criteria of the participants as presented in 2.2 participants, I would introduce or review the rational of selection family caregivers who provided care to disabled family members?

Author Response

Response to Reviewer 1 Comments

We appreciate Reviewer 1 to his/her thoughtful comments.

We have responded to the comments as follows:

  • Caregivers and family caregivers are not exactly the same, I would consistently use one of them, e.g., family caregivers, throughout the whole manuscript.

Response:

Thank you for pointing this out. “Caregiver”, “caregivers” were changed to “family caregiver”, “family caregivers” throughout the text.

  • I am confused by the description of hypothesis. It seems to me that the descripted hypothesis is a common sense in that: “having cancer is expected to be related to higher distress among caregivers.”, which may limit the novelty of the manuscript.

Response:

Thank you for your thoughts regarding the hypothesis description.

As this is just one part of the study and intuitive/common sense, we have deleted this as a hypothesis so that the study aim is only described without the hypothesis description.

However, we believe that focusing on caregiving in cancer patient is becoming very important recently. The comparison between family caregivers and the others among cancer patients has been very few in Japan, as for Kumagai mentioned some family caregiver diseases as covariates of serious distress but he did not include cancer in the result’s table, only indicating in the foot note that cancer was included as covariate and not shown in any supplementary table; ours is the first study whose main focus as exposure of distress is cancer in caregivers. We also believe that this study would also bring an important information about the psychological problem among family caregivers with cancer for clinicians, policymakers, and researchers in other countries. We described the importance in the Discussion section.

Page 8 L 251-253 (4. Discussion, 1st paragraph)

  This is the first study to illustrate the higher risk of psychological distress among family caregivers with cancer compared to those without cancer in Japan.

Page 9 L 295-300 (4. Discussion, 3rd paragraph)

  For a worldwide future implication, since Japan is considered a super-aged society with the first position as having the largest percentage of older adults (28%) in their population [45]. The country is taken as a model of research and management of aging across different fields, our study would bring an important information about the psychological problem among family caregivers with cancer for clinicians, policymakers, and researchers in other countries.

Reference

  1. United Nations, Department of Economic and Social Affairs, World Population Prospects 2019. https://population.un.org/wpp/Download/Standard/Population/ (September 10 2021).

  • I am also confused by the description of the study purpose in introduction section (page3, lines 100-102), maybe it is easier to understand as presented under abstract section: “to describe the characteristics of caregivers with cancer compared to those without and analyze the association of having cancer and caregivers’ psychological distress in Japan.”

Response:

We agree that the aim described in the abstract “to describe the characteristics of caregivers with cancer compared to those without and analyze the association of having cancer and caregivers’ psychological distress in Japan.” is more complete than that in the Introduction “this study aimed to describe the characteristics of caregivers with cancer and examine the association between being a cancer patient and distress among caregivers in Japan.” We revised the aim in the abstract as suggested.

  • Considering the criteria of the participants as presented in 2.2 participants, I would introduce or review the rational of selection family caregivers who provided care to disabled family members?

Response:

We have endeavored to provide a clearer description of the criteria.  While ageing and illness impairing function would be a form of disabled care recipient, we now provide a more adequate definition of family caregiver should be written according to your comment. The additional sentences about the definition are described as follows.

Page 3, L 119-125 (2.2. Participants)

 The participants were primary/family caregivers who provided care to aged, chronically ill or disabled family members at home with an LTCI certification. In the CSLC, the care recipients were defined as the respondent to the survey for long-term care. The CSLC also included the information of the respondent household and who provides care to the care recipients as a family member. With this information, we defined the family caregiver as someone primarily providing care to the care recipient as the study participants. This algorithm to define a family caregiver is used in previous studies using the CSLC [26, 27].

Reference:

  1. Sugiyama, T.; Tamiya, N.; Watanabe, T.; Wakui, T.; Shibayama, T.; Moriyama, Y.; Yamaoka, Y.; Noguchi, H., Association of care recipients’ care-need level with family caregiver participation in health check-ups in Japan. 2018, 18, (1), 26-32.
  2. Tokunaga, M.; Hashimoto, H.; Tamiya, N., A gap in formal long-term care use related to characteristics of caregivers and households, under the public universal system in Japan: 2001–2010. Health Policy 2015, 119, (6), 840-849.

Reviewer 2 Report

This paper shows an assessment of caregivers distinguishing between those who have cancer and those who do not. It is very interesting to see how the number of caregivers with cancer has increased over the years. It also confirms the increased distress in this population.

I have small comments to make on your manuscript:

Firstly, I think the abtract should include the size of the sample analysed.

Secondly, the surveys have been collected in different years. Could one person have answered the survey in all three years? This aspect should be clarified in the manuscript.

Finally, I would like to know why caregivers under the age of 40 were not included. Were there no younger caregivers or were they excluded from the study? In the abstract it seems that the majority group was between 40 and 64 years old, whereas in the methodology you say that only those aged 40 and over were included. This is not clear.

Thank you for allowing me to read and evaluate your manuscript.

Author Response

Response to Reviewer 2 Comments

We appreciate Reviewer 2 to his/her thoughtful comments.

We have responded to the comments as follows:

  • Firstly, I think the abtract should include the size of the sample analysed.

Response:

Thank you for pointing this out. We included the sample number in the abstract.

Page 1 L21 (Abstract)

The participants were 5,258 family caregivers aged ≥40 years, caring for only one family member whose information in the dataset was available for all the covariates included in the model.

  • Secondly, the surveys have been collected in different years. Could one person have answered the survey in all three years? This aspect should be clarified in the manuscript.

Response:

There might be a possibility of this, although it should be very rare.

We added the following explanation in the Materials and Methods section:

Page 3, L 108-110 (2.1. Data source, 1st paragraph)

  The survey has been conducted annually, whereas questionnaires for long-term care recipients have been completed every three years; we used data from 2010, 2013, and 2016; samples were extracted independently in every survey.

We added the following sentences in the Discussions (limitation) section:

Page 10, L 327-333 (4. Discussion, 6th paragraph)

Fifth, there is no way to tell if the same family caregiver-care recipient dyads took part in more than one wave of cross-sectional surveys since our study data was anonymized. The inter-person correlation should have been statistically adjusted in our analysis if the same subject participated in the survey multiple times, but we were unable to do so due to the lack of a subject identifier in our data. However, it should be very rare to participate in multiple years because each survey extracted about 2,500 unit areas from approximately 1 million unit areas throughout Japan using a stratified sampling method. [48,49]

Reference

  1. Ministry of Internal Affairs and Communications, Overview of the 2015 Census Postcensal Survey. https://www.stat.go.jp/info/kenkyu/kokusei/yusiki27/pdf/05sy0300.pdf (September 10 2021).
  2. Ministry of Health, Labour and Welfare, The Comprehensive Survey of Living Conditions, Outlines of Survey. https://www.mhlw.go.jp/toukei/list/20-21tyousa.html#anchor09 (September 10 2021).

  • Finally, I would like to know why caregivers under the age of 40 were not included. Were there no younger caregivers or were they excluded from the study? In the abstract it seems that the majority group was between 40 and 64 years old, whereas in the methodology you say that only those aged 40 and over were included. This is not clear.

Response:

Thank you for your suggestion. In the present study, we focused on the cancer status of the family caregivers. We recognize that the incidence of cancer increases drastically by aging; we needed to truncate the samples by age because younger caregivers with cancer were very few. We decided the 40 years cut-off point according to the screening baseline for cancer diagnosis in Japan, which gives us an idea of which since age group with could find a higher cancer prevalence or incidence. According to your comment, we inserted more explainable sentence for this criterion of age as follows:

Page 3, Line 127-129

  Because younger caregivers with cancer were very few, we excluded participants aged less than 40 years; the threshold was decided based on the target age for cancer screening in Japan [28].

Reference

  1. Hamashima, C., Cancer screening guidelines and policy making: 15 years of experience in cancer screening guideline development in Japan. Japanese journal of clinical oncology 2018, 48, (3), 278-286.

Round 2

Reviewer 1 Report

Thank you for your carefully executed and complete revisions.

Reviewer 2 Report

No comments. I am very grateful to read the modified version of the manuscript.